# Functional characterization of a bioengineered liver after heterotopic implantation in pigs

Brett D. Anderson [ID][1,5], Erek D. Nelson [ID][2,5], DongJin Joo[2,3], Bruce P. Amiot [ID][2], Aleksandr A. Katane[1], Alyssa Mendenhall[1], Benjamin G. Steiner[1], Aron R. Stumbras[1], Victoria L. Nelson[1], R. Noelle Palumbo[1], Thomas W. Gilbert[1], Dominique S. Davidow [ID][1], Jeffrey J. Ross [ID][1] & Scott L. Nyberg [ID][2,4✉]

Organ bioengineering offers a promising solution to the persistent shortage of donor organs. However, the progression of this technology toward clinical use has been hindered by the challenges of reconstituting a functional vascular network, directing the engraftment of specific functional cell types, and defining appropriate culture conditions to concurrently support the health and phenotypic stability of diverse cell lineages. We previously demonstrated the ability to functionally reendothelialize the vasculature of a clinically scaled decellularized liver scaffold with human umbilical vein endothelial cells (HUVECs) and to sustain continuous perfusion in a large animal recovery model. We now report a method for seeding and engrafting primary porcine hepatocytes into a bioengineered liver (BEL) scaffold previously reendothelialized with HUVECs. The resulting BELs were competent for albumin production, ammonia detoxification and urea synthesis, indicating the presence of a functional hepatocyte compartment. BELs additionally slowed ammonia accumulation during in vivo perfusion in a porcine model of surgically induced acute liver failure. Following explant of the graft, BEL parenchyma showed maintenance of canonical endothelial and hepatocyte markers. Taken together, these results support the feasibility of engineering a clinically scaled functional BEL and establish a platform for optimizing the seeding and engraftment of additional liver specific cells.

[1] Miromatrix Medical Inc, Eden Prairie, MN, USA. [2] Department of Surgery, Mayo Clinic, Rochester, MN, USA. [3] Department of Surgery, Yonsei University College of Medicine, Seoul, South Korea. [4] William J. von Liebig Center for Transplantation and Clinical Regeneration, Mayo Clinic, Rochester, MN, USA. [5] These authors contributed equally: Brett D. Anderson, Erek D. Nelson. ✉email: Nyberg.Scott@mayo.edu

Liver transplantation currently represents the only curative therapy for acute and chronic liver failure; however, access to this treatment is limited by a persistent shortage of donor organs. Whole-organ bioengineering has the potential to fulfill this unmet need by offering a virtually limitless supply of bioengineered liver grafts. One of the most promising technologies for whole-organ bioengineering is perfusion decellularization of whole porcine organs to obtain an acellular scaffold that retains the native extracellular matrix (ECM) composition and tissue microenvironments that provide a favorable environment for seeding parenchymal cells, and the native vascular networks critical to sustaining those parenchymal cells with nutrients and oxygen. The scaffolds are clinical sized for human transplant and can then be repopulated with organ-specific human cells using a perfusion-based bioreactor. First applied in a rat cardiac model[1], perfusion decellularization has since enabled the creation of acellular scaffolds from a variety of whole organs—including, heart[1], liver[2,3], lung[4], kidney[5,6], and pancreas[7,8]—all of which have demonstrated an ability to support engraftment of tissue-specific cell types.

Engraftment of multiple resident liver cell types—including hepatocytes[2,9–11], endothelial cells[10,12], and cholangiocytes[13]—into perfusion decellularized liver scaffolds has been shown in small, preclinical models and proof-of-concept attempts to describe a clinical-scale BEL in vitro[14–16]. As the majority of in vivo BEL functional studies have relied on small animal models, they do not recapitulate many of the unique challenges of translating whole-organ bioengineering to a clinically relevant scale. The scale-up, cell seeding optimization, and development of bioreactor culture conditions to support the multiple required cell types remain important hurdles in the development of a clinically translatable BEL.

We recently reported the reconstitution of a functional vascular network in a porcine whole liver scaffold with HUVECs, which reproducibly sustained in vivo perfusion for an excess of 10 days in a porcine liver transplant model under a steroid-based immunosuppression regimen.[17] Building on this success, the current study reports an optimized method for the seeding and simultaneous culture of HUVECs and primary porcine hepatocytes in decellularized porcine liver scaffolds. The resulting BEL constructs resemble native liver tissue with respect to the localized engraftment of endothelial cells and hepatocytes, detectable hepatic function during bioreactor culture, and sustained blood perfusion at physiologic pressures in an ex vivo perfusion model. The current study further shows that these BELs are amenable to surgical implantation and continuous perfusion in a large animal heterotopic liver transplant model, with the maintenance of cell viability and functional marker expression 48 h following implantation. Collectively, these studies describe a robust platform for the seeding and culture of multiple liver cell types, and demonstrate proof of concept for bioengineering a therapeutic liver construct at a clinically relevant scale.

## Results

### Characterization of BEL constructs seeded with primary endothelial cells and hepatocytes.
Freshly explanted porcine livers were decellularized by sequential perfusion with Triton X-100 and SDS detergent solutions to generate the acellular scaffolds used in these studies (Fig. 1a–c). Histological staining demonstrated complete removal of cellular material from the decellularized scaffold (Fig. 1d, e) and retention of the extracellular matrix (ECM) proteins collagen I and collagen IV (Fig. 1f–i). Decellularized livers were mounted in custom bioreactors (Fig. 1j) and perfused with antibiotic-free endothelial culture media for 72 h to confirm the sterility of the scaffold (Fig. 1k).

Human umbilical vein endothelial cells (HUVECs) were expanded in 2D tissue culture flasks and seeded into decellularized BEL scaffolds through the suprahepatic inferior vena cava (sIVC), followed by the portal vein (PV) 24 h later (Fig. 1k, i). We previously reported that daily glucose consumption rates (GCR) provide a robust, non-invasive metric for monitoring cell proliferation in HUVEC-seeded BEL constructs and is predictive of successful perfusion outcomes following in vivo inplantation[17]. BELs were cultured until a minimum GCR of 50 mg/h was observed (typically 13–16 days following HUVEC seeding) prior to infusing hepatocytes into the scaffold (Fig. 1k, m).

Primary hepatocytes were isolated from freshly explanted porcine livers and enriched by differential sedimentation through multiple low-speed centrifugation and washing steps to achieve typical hepatocyte purity of 85-95% as measured by albumin-positive flow cytometry (Supplementary Fig. S1). Hepatocytes were seeded through the bile duct and cultured for an additional 48 h under continuous PV perfusion with media formulated for the simultaneous culture of endothelial cell and hepatocytes (referred to herein as co-culture media; see "Methods"). Hematoxylin and eosin (H&E), as well as immunofluorescence staining for hepatocyte-specific markers (albumin and fumarylacetoacetate [FAH]) 48 h post hepatocyte seeding revealed a clustered distribution of hepatocytes throughout the scaffold parenchyma (Fig. 2a–d), consistent with prior studies that utilized the bile duct for cell infusion[9,10,13,18]. As we previously reported[17], and similar to native liver tissue, endothelial cells localized within large vessels expressed high levels of CD31, while endothelial cells localized within parenchymal capillaries had increased expression of the liver sinusoidal endothelial cell (LSEC) marker LYVE1 and reduced expression of CD31 (Fig. 2e).

To characterize the in vitro function of BELs over the course of bioreactor culture, levels of cell-derived soluble factors were quantified in culture media samples from scaffolds seeded with both HUVECs and hepatocytes (co-culture), as well as scaffolds that received only one cell type (HUVEC only and hepatocyte only) (Fig. 2f). BELs seeded with HUVECs (HUVEC only and co-culture) exhibited increasing production of the endothelial cell-derived von Willebrand factor (vWF) over time (Fig. 2g). Importantly, vWF production in co-culture BELs following hepatocyte seeding was similar to levels observed in HUVEC only BEL controls, suggesting that the addition of hepatocytes to the scaffold did not compromise endothelial cell viability and function. Albumin levels were quantified in media samples from co-culture and hepatocyte only BELs, and there was no appreciable difference in albumin production observed, suggesting that hepatocyte function not impacted by endothelial cells in the scaffold. (Fig. 2h). To further assess key metabolic functions of hepatocytes in BEL scaffolds, ammonia detoxification and urea production assays were adapted to bioreactor scale culture based on prior 2D culture and hepatocyte spheroid studies[19]. In brief, BELs were challenged with fresh co-culture media supplemented with 0.8 mM ammonium chloride 16–20 h following hepatocyte seeding and samples were taken at intervals up to 23 h after the challenge for quantification of ammonia and urea (Fig. 2i). As expected, HUVEC only BELs showed a steady increase in ammonia levels over time due to ongoing cellular metabolism. In contrast, hepatocyte only and co-culture BELs both showed substantial ammonia clearance with similar clearance kinetics although hepatocyte only BELs showed slightly more ammonia clearance (Fig. 2j). Hepatocyte only and co-culture BELs showed similar rates of urea release, while HUVEC only BELs showed only a slight increase in urea compared to acellular graft controls (Fig. 2k). These assays confirm the ability of hepatocytes to retain their function in high-density cultures in a decellularized liver scaffold. In addition, the data show that endothelial cells cultured

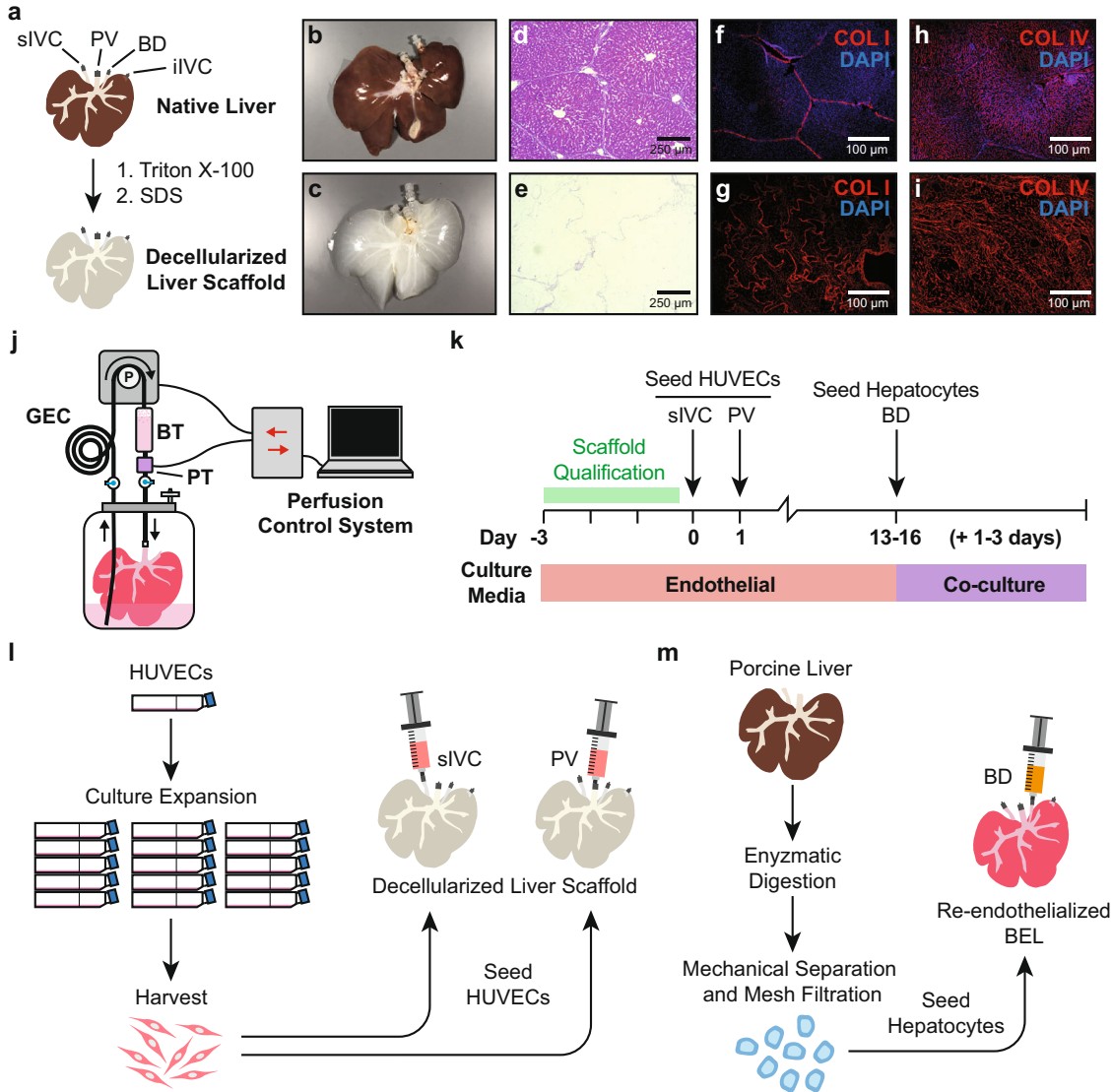

**Fig. 1 Overview of approach to bioengineering a porcine-derived whole liver construct. a** Overview of vessel cannulation and detergent perfusion methods for whole liver decellularization. **b**, **c** Photographs of representative porcine (**b**) native and (**c**) decellularized whole livers. **d**, **e** Hematoxylin and eosin staining of histological sections from (**d**) native and (**e**) decellularized porcine liver tissue demonstrating the efficient removal of cellular material while preserving the native tissue architecture. **f–i** Representative immunofluorescence micrographs from native (**f**, **h**) and decellularized (**g**, **i**) liver tissue demonstrating retention of Collagen I and Collagen IV, respectively, in cell-free scaffolds. The absence of DAPI staining in the decellularized scaffolds demonstrates removal of cellular DNA. **j** Schematic of perfusion bioreactor design. BELs are suspended in a vessel with culture media. An extended coil of silicone tubing (GEC) is included in the perfusion circuit to facilitate gas exchange. A bubble trap is positioned directly upstream of the BEL perfusion inlet. During media perfusion, a pressure transducer provides real-time feedback to a controller which in turn adjusts the flow rate on a peristaltic pump to maintain a constant perfusion pressure. **k** Schematic depicting bioreactor culture and cell seeding events. Decellularized scaffolds were pre-qualified in antibiotic-free media for 3 days prior to HUVEC seedings. HUVEC-seeded liver constructs were cultured in endothelial cell growth media until a 24 h average glucose consumption rate of 50–90 mg/h was reached (typically 13–16 days), at which point hepatocytes were seeded into the scaffold. BELs were maintained in co-culture media formulated for simultaneous culture of endothelial cells and hepatocytes for the remainder of the experiment (typically 1–3 additional days). **l** Overview of HUVEC culture and seeding approach to generate a reendothelialized liver construct. Cells were infused first through the cannulated iIVC followed by a second cell infusion through the PV 24 h later. **m** Schematic of porcine donor liver hepatocyte isolation and seeding of bioengineered liver constructs. Whole porcine livers are enzymatically digested, and dissociated cells were filtered through a series of mesh sieves to remove large debris and cell aggregates. Hepatocytes were enriched through multiple low speed (70 × g) centrifugation and washing steps. DAPI—4′,6′-diamidino-2-phenylindole; BEL—bioengineered liver; GEC—gas exchange coil; BT—bubble trap; iIVC—infrahepatic inferior vena cava; PV—portal vein.

in the scaffold did not impede the functionality of hepatocytes in the scaffold in vitro.

**Acute blood perfusion studies to assess vascular patency of BELs.** The ability to sustain blood perfusion in vivo is essential to the successful development of any bioengineered organ. To determine the ability of BELs in this current study to sustain

short-term perfusion with blood, an ex vivo perfusion circuit was created to recirculate porcine blood through BELs (seeded and cultured as above; [Fig. 2f]) at a regulated target pressure of 12 mmHg while monitoring flow rates over 30 minutes (Fig. 3a). To mimic the initial graft reperfusion rates experienced in a surgical liver transplant model, initial blood flow through the BELs was set to 350 mL/min to fully inflate the graft, and flow

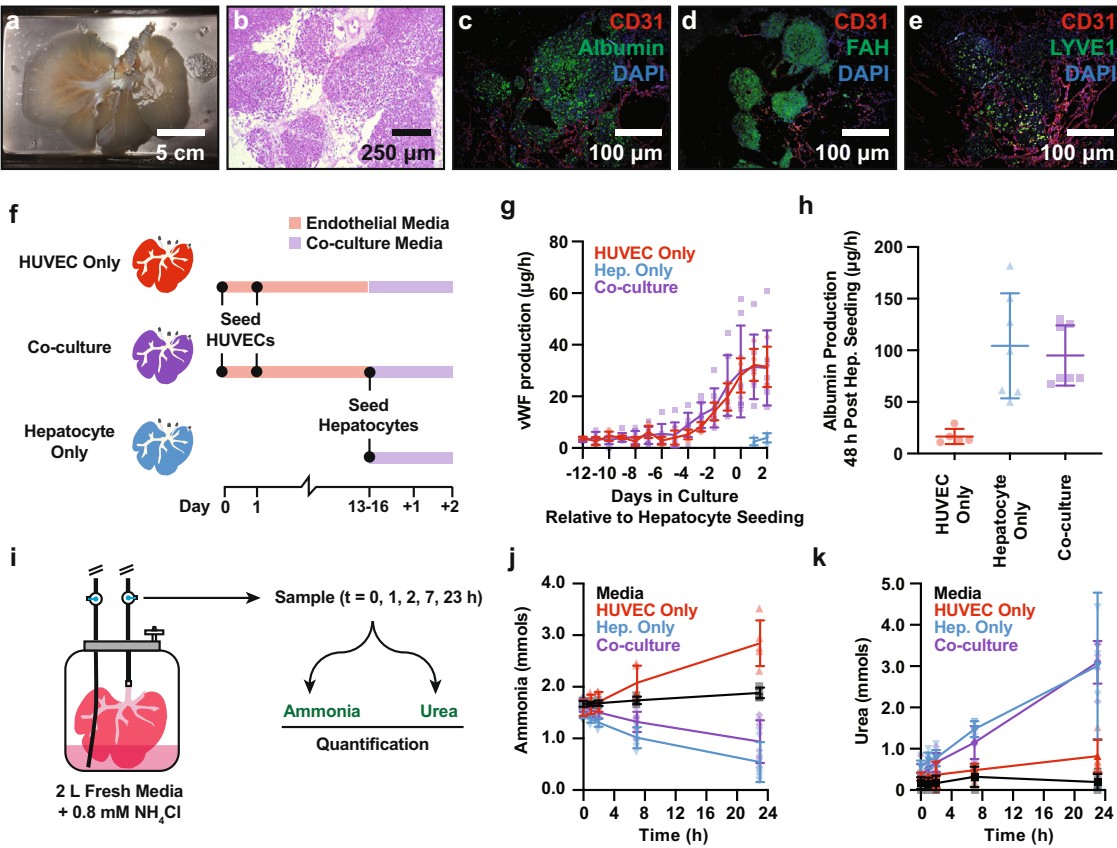

**Fig. 2 Histological and functional characterization of BEL constructs. a** Representative photograph of a BEL seeded with HUVECs and porcine hepatocytes. **b** Hematoxylin and eosin staining of representative co-culture BEL tissue sections fixed 48 h after seeding hepatocytes. **c-e** Immunofluorescent staining of cell lineage markers in non-serial tissue sections 48 h after seeding hepatocytes: (**c**) CD31 & albumin; (**d**) CD31 & FAH; (**e**) CD31 and LYVE1. **f** Schematic depicting seeding and culture timeline for HUVEC only, hepatocyte only, and co-culture BEL constructs used in (**g, h, j, k**). **g** vWF production in grafts before and after hepatocyte seeding. Data from independent HUVEC only ($n = 5$), hepatocyte only ($n = 7$), and co-culture ($n = 7$) BEL constructs are shown. Error bars denote the mean and standard deviation at each time point. **h** 24-h average albumin production in co-culture grafts 48 h following hepatocyte seeding. Data from independent HUVEC only ($n = 5$), hepatocyte only ($n = 7$), and co-culture ($n = 7$) BELs are shown. Error bars denote the mean and standard deviation. **i** Schematic of in vitro ammonia clearance and urea production assay. Ammonium chloride is added to the bioreactor media at a concentration of 0.8 mM. Ammonia and urea levels are measured in media samples taken at $t = 0, 1, 2$ 7, and 23 h following the addition of ammonium chloride. Error bars denote the mean and standard deviation each time point. **j, k** Ammonia clearance (**j**) and urea production kinetics (**k**) following the addition of ammonium chloride to the bioreactor perfusion media. Data from independent HUVEC only ($n = 5$), hepatocyte only ($n = 7$), and co-culture ($n = 7$) BELs, media only controls ($n = 4$) are shown. Error bars denote the mean and standard deviation each time point. BEL—bioengineered liver; FAH—fumarylacetoacetate hydrolase; CD31—cluster of differentiation 31; LYVE1—lymphatic vessel endothelial hyaluronan receptor 1; vWF—von Willebrand factor.

rates were subsequently allowed to automatically correct to target a perfusion pressure of 12 mmHg. Additionally, prior to BEL perfusion, the activated clotting time (ACT) of heparinized porcine blood (ACT > 1500) was titrated to a slightly elevated, physiologically relevant range of 170–230 (just high enough to inhibit clotting within the in vitro blood circuit components) through the addition of protamine sulfate. To establish benchmarks for optimal perfusion as well as scaffold thrombosis in this system, freshly explanted porcine livers ($n = 2$) and unseeded decellularized liver scaffolds ($n = 2$) were subjected to continuous perfusion in the blood circuit. As expected, native organs exhibited relatively stable flow rates in excess of 300 mL/min by the end of 30 min, whereas decellularized scaffolds thrombosed within 5 min (Fig. 3b, c). Four out of five HUVEC only BELs exhibited relatively stable flow rates (250-350 mL/min) over 30 minutes (similar to freshly explanted livers), while one lost flow gradually over the course of perfusion (Fig. 3b, c). Similar to the decellularized scaffolds, hepatocyte only BELs ($n = 3$) experienced a rapid loss in flow within the first five minutes,

(Fig. 3b, c), demonstrating the importance of endothelial cells in sustaining perfusion through the scaffold. Co-culture BELs ($n = 5$) exhibited intermediate flow rate profiles (Fig. 3b, c), presumably due to increased resistance within the scaffold vasculature from the hepatocyte seeding. Nevertheless, all co-culture BELs still maintained blood flow (118–293 mL/min) after 30 min of continuous perfusion at physiologic pressures.

To further characterize the vascular blood flow through liver scaffolds seeded with both HUVECs and hepatocytes, a porcine ex vivo blood perfusion model was employed whereby the PV, sIVC and infrahepatic IVC (iIVC) of anesthetized pigs (80–100 kgs) were cannulated and used to establish a perfusion circuit through co-culture grafts under physiologic venous flow (Fig. 3d). Animals were heparinized to an ACT of ~250 to prevent clot formation in the cannula or tubing. Flow was measured at 150–250 mL/min at 30 min and real-time angiography further confirmed continuous perfusion and revealed vascular flow including visualization of capillary beds in the majority of the co-culture BEL (Fig. 3e, Supplementary Movie S1).

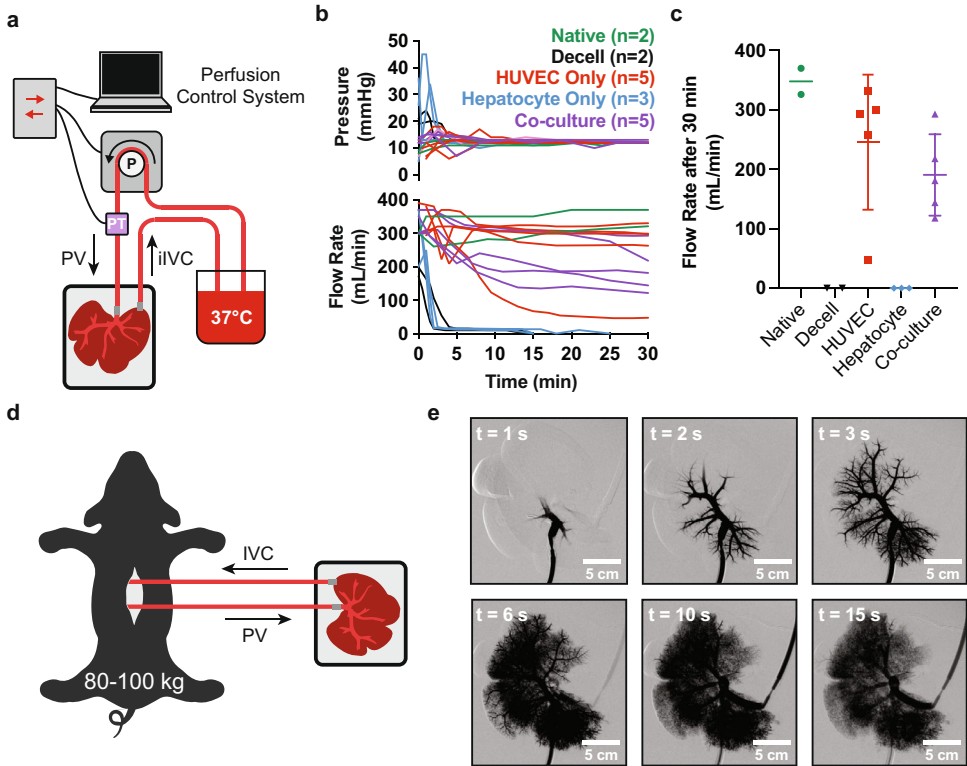

**Fig. 3 Acute blood perfusion studies to assess vascular patency in BELs. a** Schematic of in vitro blood perfusion circuit. 37 °C porcine blood is perfused at 12 mmHg through the PV with a peristaltic pump and returned to a reservoir through the IVC. **b** Summary plots of pressures and flow rates measured over 60 minutes during in vitro blood perfusion studies using HUVEC only, hepatocyte only, or co-culture BELs. Freshly explanted (Native) porcine livers and decellularized scaffolds (Decell) were included as benchmarks for idealized perfusion and rapid thrombosis, respectively. **c** Violin plots summarizing BEL flow rates from (**b**) after 30 min of perfusion. **d** Schematic of ex vivo blood perfusion model. A synthetic perfusion circuit is established by cannulating the PV and sIVC within an anesthetized pig. Blood flow is diverted from the animal's cannulated PV to the BEL PV and returned from the BEL sIVC into the animal's cannulated iIVC. **e** Real-time angiography time lapse imaging following contrast infusion. Imaging was performed after 30 min of continuous blood perfusion. PV—portal vein; BEL—bioengineered liver; sIVC—suprahepatic inferior vena cava, iIVC—infrahepatic inferior vena cava.

**Implantation of co-culture BELs in a large animal model.** To assess the ability of co-culture BELs to sustain 48 h of continuous physiologic perfusion and hepatocyte function in vivo, a porcine heterotopic liver transplant model of acute liver failure was employed. BELs (48 h post hepatocyte seeding) were situated in a heterotopic position in the hepatorenal space and end-to-side anastomoses were performed between the BEL PV to the native PV and BEL sIVC to native iIVC, respectively (Fig. 4a, Supplementary Figs. S4, 5, Supplementary Movie S2). Total portal vein flow was diverted into the BEL by ligating and dividing the native PV between the BEL and native liver. The implanted BEL (260 ± 40.8 g) was ~30% the native liver size resulting in a small-for-size syndrome. To reduce portal hypertension while maintaining sufficient flow through the implanted BEL, a small portocaval shunt was created utilizing a 4 mm ringed polytetrafluoroethylene (PTFE) graft which was anastomosed end-to-side between the recipient's PV and iIVC. Blood flow through the portocaval shunt was measured at between 20 and 50 mL/min, or ~10% of the flow through the BEL, after anastomosis and titrated as necessary via banding or ligation of the portocaval shunt.

Following BEL perfusion, the native liver was isolated from arterial perfusion by ligating and dividing all structures within the hepatoduodenal ligament including the common bile duct and the hepatic artery complex. The caudate lobe was devascularized via oversewing and compression with locking suture until cut caudate parenchyma was cyanotic and did not demonstrate bleeding from cut edges. Flow through the BEL the first few hours after perfusion

was 120–410 mL/min as measured by a Transonic flow probe prior to closing the animal. Computed tomography (CT) was utilized post-operatively to confirm BEL perfusion and loss of flow to the native liver (Fig. 4b, c). Follow up CT imaging at Day 1 and Day 2 confirmed that BEL perfusion was sustained throughout the remaining duration of the acute implant studies (Fig. 4c).

**Functional characterization of BELs in Vivo.** In addition to showing sustained perfusion of a hepatocyte/endothelial BEL in vivo, the survival and functionality of hepatocytes were assessed. Previous studies have correlated the relationship between intracranial pressure (ICP) and blood ammonia levels during acute liver failure[19]. Therefore, a totally internal intracranial probe (NEUROVENT-P-tel, Raumedic, Mills River, NC) was implanted 5 days prior to surgery to allow monitoring of the intracranial pressure (ICP). Termination of the study was performed when an animal experienced (1) ICP of 20 mmHg or greater for 2 h, (2) mean arterial pressure (MAP) was 30 or less for 2 h, or (3) 48 h of survival was achieved. Control group animals ($n = 2$) underwent ICP probe insertion and subsequent creation of a total portocaval shunt by directly anastomosing native portal vein to native iIVC before devascularization of the native liver as described above. This resulted in termination at 24 and 48 h (Fig. 4d). The 24-h survivor was terminated due to a MAP less than 30, at which point the ICP was 17. The 48-h survivor had a MAP of 42 and ICP of 16 at the time of termination. Blood ammonia levels in the control animals climbed

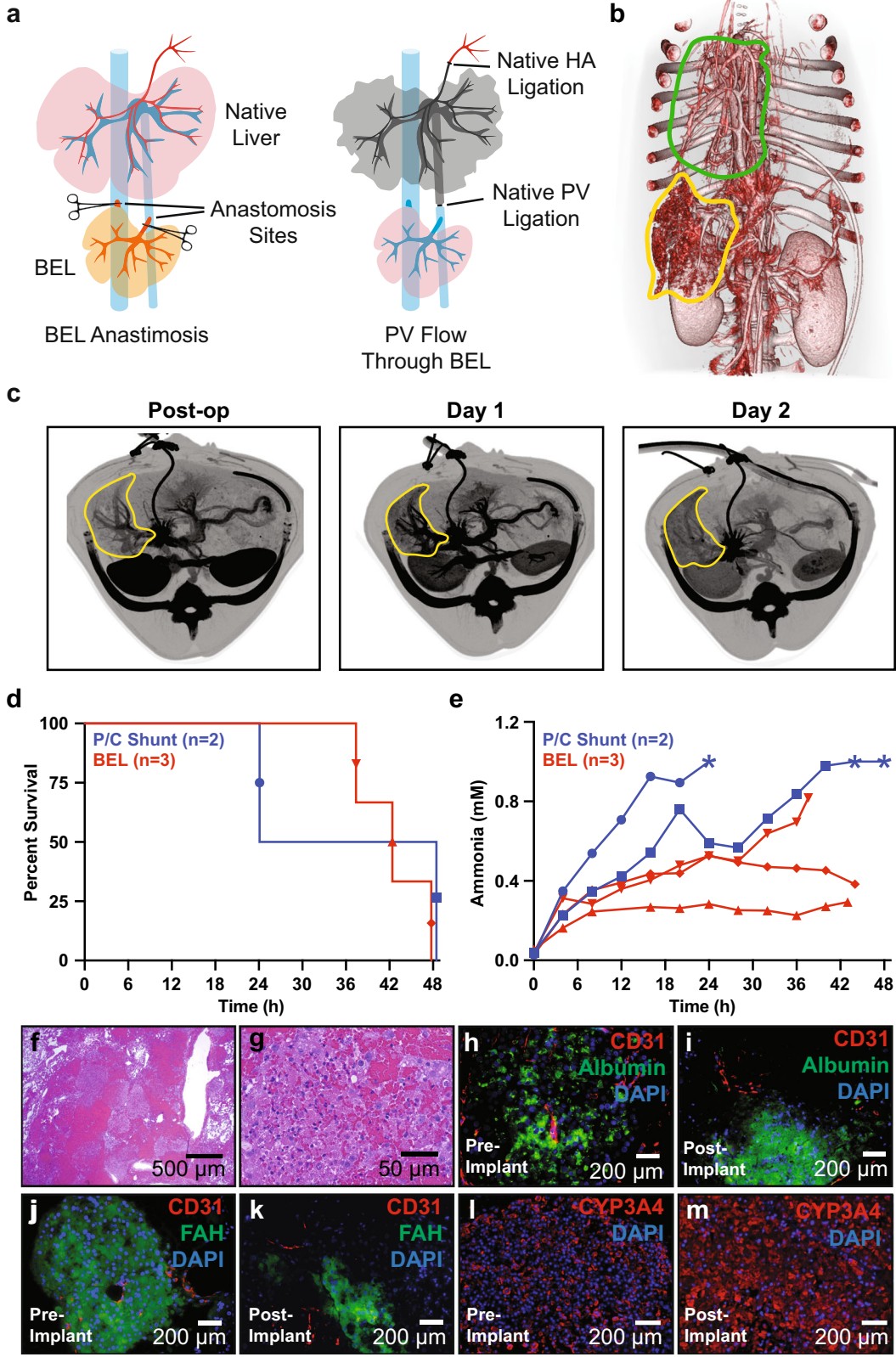

quickly with levels surpassing 0.5 mM within 15 h of surgery (Fig. 4e). In these control animals, there was no ascites accumulation indicative of negligible portal hypertension as expected with the portocaval shunt.

In contrast, the experimental group animals (*n* = 3), having received co-culture BEL implants, were terminated at 37 h, 42 h,

and 48 h (Fig. 4d). The 37-h survivor showed an ICP greater than 20 mmHg with a MAP of 37 mmHg. The 42-h survivor had a low MAP secondary to respiratory failure (diagnosed with acute pulmonary edema and large bilateral pleural effusions on CT) with an ICP of 11. The 48-h survivor reached the end of the study and had a MAP of 38 and ICP of 17. The BEL group also showed

**Fig. 4 Heterotopic implantation of co-culture BELs in a large animal model. a** Schematic of heterotopic BEL implant surgical model. See text for details. **b** Post-operative 3D-reconstruction from CT imaging demonstrating BEL perfusion and devascularization of native liver. BEL is outlined in yellow and native liver is outlined in green. **c** Representative axial CT imaging of recipient animal at post-op, 24 h, and 48 h time points. BEL is outlined in yellow. **d** Kaplan–Meier curves showing animal survival times within portocaval shunt and BEL implant groups. Symbols are matched to ammonia values in (**d**). **e** Post-operative blood ammonia levels measured in BEL implant recipient animals ($n = 3$) and portocaval shunt animals ($n = 2$) over the duration of the experiment. Asterisks (*) denote data points that were above the upper limit of quantification of the assay (1 mM). **f, g** Representative histological section of BEL tissue explanted 48 h post-implant showing viable hepatocytes and endothelialized vasculature. **h, i** Representative immunostaining BEL tissue (**h**) pre-implant and (**i**) explanted 48 h post-implant showing maintenance of CD31 and albumin expression. **j, k** Representative immunostaining BEL tissue (**j**) pre-implant and (**k**) explanted 48 h post-implant showing maintenance of CD31 and FAH expression. **l, m** Representative immunostaining BEL tissue (**l**) pre-implant and (**m**) explanted 48 h post-implant showing maintenance of CYP3A4 expression. HA—hepatic artery; BEL—bioengineered liver; P/C—portocaval; CT—computed tomography; FAH —fumarylacetoacetate hydrolase; CYP3A4—cytochrome p450 3A4.

an increase in blood ammonia levels to approximately 0.5 mM, but leveled off or decreased, except for one graft that lost graft perfusion (37-h survivor) that resulted in a rapid increase in ammonia and ICP that prompted termination at 37 h (Fig. 4e). The ammonia data are suggestive of hepatocyte function and are consistent with the observed ex vivo ammonia clearance. However, interpretation of the results of the heterotopic implant model may be confounded by the differential portal-systemic pressure between models.

As discussed above, the co-culture BELs had reduced in vitro blood flow at a controlled pressure when compared to HUVEC only BELs. Combined with the expected small-for-size syndrome of the model and despite the attempt to regulate portal hypertension with a small portosystemic shunt, the presence of the increased resistance within the co-culture BEL resulted in increased portal hypertension compared with the control group's portocaval shunt as evidenced by the increased ascites output in the experimental group compared to the control group (Supplementary Fig. S3). The presence of portal hypertension within the experimental group, but not the control group, could result in increased ammonia production secondary to mesenteric venous congestion and underestimation of the ammonia clearance function in the experimental group. HUVEC only BELs demonstrated mixed results. Based on in vitro flow data, this group experienced less portal hypertension than the co-culture BEL group, but significantly more than the control portocaval shunt group. This is also supported by the accumulation of ascites in the HUVEC only BEL group, but not to the extent demonstrated by the co-culture group. Taking these factors together, we concluded that portal hypertension is an important consideration in BEL development as it may limit the hepatocyte mass that can be seeded per unit volume of the graft. Therefore, an increased graft size may be needed to increase the hepatocyte mass while reducing the portal hypertension to decrease ammonia production secondary to mesenteric venous congestion and provide a better estimation of graft function. BELs retained their shape and size (Supplementary Fig. S6), histological examination of explanted BELs with Hematoxylin & Eosin (H&E) (Fig. 4f, g) and immunostaining for albumin, FAH, CD31, CYP3A4, and LYVE1 (Fig. 4h–m, Supplementary S7, Supplementary S8)) pre- and post-implant showed viable porcine hepatocytes and HUVECs throughout the explanted BEL suggesting that seeded cells remained viable and phenotypically stable throughout the duration of in vivo perfusion.

## Discussion
Despite recent advances in tissue engineering technologies, the ability to engineer whole-organ systems at a clinically relevant scale has remained extremely challenging, largely due to the inherent difficulties in fabricating scaffolds that provide suitable vascular beds to oxygenate and nourish seeded cells and to

adequately recapitulate the complex microenvironments and functions of native tissues. Perfusion decellularization offers an elegant solution to this problem by enabling the creation of human-scaled whole-organ scaffolds that preserve the integrity of vascular networks and parenchymal microenvironments that promote robust engraftment and functionality of tissue-specific cell types.

The first step towards realizing a clinically relevant BEL is to ensure long-term patent vascularization of the graft. Multiple groups have attempted to address this challenge either by seeding a decellularized liver graft with endothelial cells[2,12,14] or by modifying native vasculature to inhibit coagulation[11,16]. However, in vivo validation of these approaches has been limited to implantation in rats, and human-scale BELs have been limited to in vitro characterization. We recently reported the feasibility of using HUVECs to repopulate the vascular matrix of a decellularized porcine liver and achieved up to 20 days of continuous perfusion without systemic anticoagulation in a porcine heterotopic implant model. We reported on the ability of seeded HUVECs to upregulate sinusoidal endothelial markers and evidence of fenestration formation. We found that the duration of patency was limited by the porcine xenogeneic rejection of human endothelial cells[17]. Importantly, we established glucose consumption rate as a reliable, non-destructive measure to predict patency of a BEL and a critical step in-process quality measure for BEL candidates for implantation. These observations mark important milestones towards the realization of a clinically relevant BEL, and other engineered organs such as kidney, lung, and heart.

Having shown durable patency of BEL vasculature, the next major milestone was to investigate any impact of the introduction of hepatocytes into the BEL on graft patency and the function of the hepatocytes following seeding and implantation. Two previous studies[2,11] in rats have shown early indicators of hepatic function after seeding rat hepatocytes into a decellularized rat liver matrix and implanting the resulting graft in a heterotopic position. Uygun et al. demonstrated the persistence of hepatocyte-specific markers by immunohistochemical staining, but only evaluated hepatocyte function after 8 h in a renal implant site[2]. Bao et al. showed decreased blood ammonia levels as compared to a 90% hepatectomy, but long-term patency required chemical modification of the vasculature[11]. Our previous study showed perfusion of a BEL seeded with only HUVECs for up to 14 days in pigs, and that duration was limited by a xenogeneic immune response to the HUVECs. The current study shows that the addition of hepatocytes to a clinically scaled BEL does not impede perfusion at physiologic pressures in vitro and in vivo. Furthermore, engraftment of hepatocytes in the parenchymal compartment reconstituted hepatic function within the BEL construct as measured by albumin secretion, urea production, and ammonia clearance in vitro, and the initial indication of ammonia detoxification activity in vivo. Histological examination

of co-culture BELs revealed that hepatocytes were appropriately localized in the parenchymal compartment, while HUVECs localized primarily within large vessels and capillary vasculature. Both cell types expressed canonical functional markers, and as previously demonstrated[17], HUVECs localized within sinusoidal capillaries acquired expression of LYVE1, suggesting a level of microenvironment-driven reprogramming toward an LSEC-like phenotype. These results support growing confidence in the ability to bioengineer a clinically sized liver construct capable of revascularization and surgical implantation with sustained perfusion and hepatocyte survival in a pig model. This represents a major step toward bioengineering transplantable livers for clinical use.

While the current study demonstrates the ability to heterotopically implant a BEL with sustained portal flow, hepatocyte survival and initial indications of in vivo function, there are multiple limitations of the present study we attempted to minimize. First, in this heterotopic large animal implantation model, the BEL graft size was limited by the space available in the abdominal cavity while the devascularized native liver was present. This limited our ability to use BELs that were more than one-third of the average native recipient liver weight and limiting the total number of seeded hepatocytes for functional assessment. Portal hypertension was consistently observed due to both small-for-size syndrome and the presence of hepatocytes within the graft which we showed to increase vascular resistance. In this model, elimination of blood perfusion of the native liver is achieved through ligation and division of the portal vein and hepatic arteries and oversewing the caudate lobe. The hilar structures and hepatogastric ligament were ligated and divided to eliminate all inflow including aberrant vascular anatomy. No collateral branches were detected on CT, but backflow through the hepatic vein complex was present although remnant liver function based on systemic venous backflow is likely to be negligible. Finally, this model keeps the devascularized native liver in situ (Supplementary Fig. S9). The dying organ may be detrimental as it is a source of systemic inflammation in a largely anhepatic animal, but also beneficial in that it is a known driver of hepatocyte proliferation and is contemporarily used clinically as such.

Our results provide a justification for pursuing the development of a larger BEL for use in an orthotopic implant model not only to show increased hepatic function, but also to demonstrate that it is possible to support long-term survival of a large animal. Based upon the present study, the continued development and optimization of a fully functional BEL is warranted. Future work will include increasing the size of the BEL to better match the native liver and increase the hepatocyte seeding concentration to enable survival after orthoptic implantation. The use of a larger graft will remedy two limitations of the current model: low mass of transplanted hepatocytes and presence of significant portal hypertension. Orthotopic liver transplantation cannot be performed by the piggy-back technique in swine since a surgical plane of separation does not exist between the pig's IVC and parenchyma of its liver, as occurs in humans. Therefore, our current studies to optimize a protocol for orthotopic placement of a BEL in a swine model will use the caval interposition technique of liver transplantation as originally proposed by Calne. In addition, development of a functional biliary tree will be required for a clinically viable BEL. The current results provide confidence in the promise of tissue engineering to provide an alternate supply of donor liver grafts for patients suffering from liver failure.

## Methods

**Decellularization of porcine livers**. Whole livers (250–350 g) were explanted from cadaveric pigs. The portal vein, sIVC and iIVC, and common bile duct (BD) were cannulated and flushed with sterile saline. Cannulated livers were decellularized by peristaltic pump (Cole Palmer, 7575-30; 77200-60)-driven vascular perfusion with 1% Triton X-100 followed by 0.6% sodium dodecyl sulfate. Solution flow rates were automatically regulated by a custom perfusion control system designed to maintain perfusion pressures between 8 and 12 mmHg. Decellularized livers were subsequently disinfected with 1000 ppm peracetic acid, washed with phosphate buffered saline, and stored at 4 °C. All aspects of the decellularization process were performed in an ISO 7 cleanroom facility.

**HUVEC cell culture and seeding of decellularized liver scaffolds**. Human umbilical vein endothelial cells (HUVECs) (Lonza, C2517A) were cultured at 37 °C and 5% $CO_2$ in antibiotic-free Endothelial Cell Growth Media (R&D Systems, CCM027) supplemented with 2% fetal bovine serum (Corning), 50 mg/L ascorbic acid (Sigma), 1 mg/L hydrocortisone (Sigma), 20 μg/L FGF (R&D Systems), 5 μg/L VEGF (R&D Systems), 5 μg/L EGF (R&D Systems), 15 μg/L $R^3$ IGF (Sigma), 1000 U/L heparin (Sigma), and 1.5 μM acetic acid (Sigma). Cells were harvested with 0.25% trypsin-EDTA (Thermo, 25200056) at 90–100% confluency. Decellularized porcine livers were mounted in bioreactors and perfused with antibiotic-free cell culture media (37 °C, 5% $CO_2$) for 72 h to confirm the absence of microbial contamination. HUVECs collected at passage 5–9 were infused through the sIVC with a syringe ($1.2\times10^8$ cells in 150 mL culture media). Following 1 h of static culture to allow for cell attachment within the scaffold, culture media supplemented with Penicillin and Streptomycin was perfused through the sIVC at 12 mmHg. Twenty-four h later, a second inoculum of HUVECs was collected and infused through the PV in the same manner as above. Following seeding, culture media was replaced daily, and volumes were continually adjusted to ensure that glucose levels remained above 0.3 g/L within a 24 h period. Media perfusion into the scaffold was maintained a pressure of at 12 mmHg during culture.

**Porcine hepatocyte isolation and seeding of BEL scaffolds**. Freshly harvested whole livers (400–600 g) were cannulated through the PV, sIVC, and iIVC, and perfused with 5 L of HBSS (Fisher, MT21022CM) to remove residual blood from the organ followed by 1 L of cold HTK solution (Essential Pharmaceuticals, 25767-735-24) to minimize ischemic injury to organs during transportation. Livers were then perfused through the PV (500–600 mL/min) with 5 L of HBSS supplemented with 2.5 mM EGTA (Sigma, E3889), allowing the first 1 L to drain to waste, and recirculating the remaining volume for 20 min. Livers were subsequently perfused with 2 L of solution comprised of 142 mM NaCl, 6.7 mM KCl, 10 mM HEPES, 5 mM N-acetyl-L-cysteine, and 1% Penicillin-Streptomycin (Sigma, P4333). Digestion was initiated with perfusion of 4 L of L-15 media (Fisher, 21083027) supplemented with 100 mg of Liberase TM (Sigma, 5401127001) and 5 mM $CaCl_2$, allowing the first 500 mL to drain to waste and recirculating the remaining volume until livers were soft with visible breakdown of the capsule (20–30 min). After digestion, 1 L of cold Williams E media (Fisher, RR090071P1) supplemented with 10% FBS (VWR, 97068-085) was poured over the livers and the capsule was gently pulled apart to release the cell suspension. To eliminate any remaining undigested tissue, cells were filtered through an 8″ wide mesh strainer, followed by a series of mesh sieves (250 μm (VWR, 57334-466), 125 μm (VWR, 57334-474), 70 μm (Fisher, NCO446099). The filtered cell suspension was brought to a final volume of 2 L with Williams E media supplemented with 10% FBS. Hepatocytes were enriched by low-speed centrifugation (70 × g, 4 °C, 10 min) and washed twice in cold William's E + 10% FBS. Cell viability and yield were quantified by trypan blue dye exclusion on a hemocytometer.

Following isolation, $2\times10^9$ porcine hepatocytes were diluted in 2 L ($1\times10^6$ cells/mL) of co-culture media (Williams' E medium (Gibco) supplemented with 1.5% fetal bovine serum (Corning), 50 mg/L ascorbic acid (Sigma), 1 mg/L hydrocortisone (Sigma), 20 μg/L FGF (R&D Systems), 5 μg/L VEGF (R&D Systems), 5 μg/L EGF (R&D Systems), 15 μg/L $R^3$ IGF (Sigma), 1000 U/L heparin (Sigma), 1.5 μM acetic acid (Sigma), 2 mL/L human insulin (Novolin), 3 g/L human albumin (CSL Behring), 150 μg/L linoleic acid (Sigma), 0.1 μM dexamethasone (Sigma), 40 ug/L human glucagon (Novaplus), 6 mg/L human transferrin (Sigma), 20 ug/L Gly-His-Lys (Sigma), 0.1 μM copper sulfate, 30 nM sodium selenite, 50 pM zinc sulfate, 1 g/L L-carnitine (Sigma), 0.2 g/L L-arginine (Sigma), and 10 mg/L glycine (Sigma). Hepatocytes were infused through the bile duct of reendothelialized BEL scaffolds (typically 13–16 days following the first HUVEC seeding) with a peristaltic pump at a rate of 50 mL/min. Hepatocyte-seeded BELs were then returned to continuous media perfusion through the PV with co-culture media at a pressure of 12 mmHg.

**Flow cytometry**. Hepatocyte purity post enrichment was quantified by intracellular anti-pig albumin staining (Bethyl A100-110A, 1:200) facilitated by detection with an Alexa Fluor 488 conjugated secondary antibody (Abcam 150129) following fixation with 4% paraformaldehyde and permeabilization in 0.1% Triton X100. Flow cytometry analysis was performed on a BD Accuri C6 Plus instrument and the resulting data were analyzed using FlowJo 10.

**Histological analysis**. Tissue samples analyzed in this study were perfused with PBS and fixed with 10% Neutral Buffered Formalin (VWR, 16004-128). Fixed tissues were paraffin embedded, sectioned, and stained using standard histological techniques. Immunofluorescence slides were deparaffinized, rehydrated and

retrieval was performed in citrate buffer, pH 6.0 (Abcam AB93678) in a programmable decloaker (Biocare, DC2012). Slides were permeabilized with PBS + 0.05% Tween-20 (Sigma, P9416) and blocked with Sea Block (ThermoFisher, 37527). Primary antibodies used were Rabbit anti-Collagen I (Abcam, AB34710, 1:100 dilution), Rabbit anti-Collagen IV (Abcam, AB6586, 1:100 dilution), Mouse anti-CD31 (Abcam, AB187377, 1:100 dilution), Rabbit anti-Albumin (Abcam, AB79960, 1:150 dilution), Rabbit anti-FAH (Abcam, AB83770, 1:100 dilution), Rabbit anti-Cytochrome P450 3A4 (Abcam, AB3572, 1:100 dilution) and Rabbit anti-LYVE1 (Abcam, AB33682, 1:100 dilution). Secondary antibodies were Goat anti-Mouse Alexa Fluor 488 (ThermoFisher, A11029, 1:500 dilution) and Goat anti-Rabbit Alexa Fluor 555 (ThermoFisher, A21429, 1:500 dilution). All antibodies were diluted in Sea Block. Slides were stained with DAPI (ThermoFisher, D1306) and mounted using ProLong Antifade Mountant (Thermo, P36961). H&E and immunofluorescence microscopy was performed on an Accuscope 3012 (H&E) and Zeiss Axioskop 40, respectively.

**Analysis of cellular metabolites and secreted factors during bioreactor culture.** Media samples from bioreactors were collected daily and assayed immediately on a CEDEX BioHT analyzer (Roche) to determine levels of glucose, ammonia, and lactate dehydrogenase activity in the culture media. Measured glucose concentrations were used to calculate daily consumption rates over a 24 h period prior to replenishing bioreactors with fresh media. A separate aliquot of each daily media sample was stored at −80 °C and thawed at the end of each experiment for quantification of soluble vWF (ThermoFisher, EHVWF) and albumin (Bethyl Laboratories, A100-110A) by ELISA.

**BEL ammonia clearance kinetics and urea production assays.** Sixteen to twenty hours after seeding hepatocytes, culture media was removed from bioreactors and 2 L of co-culture media supplemented with 0.8 mM ammonium chloride. Bioreactor media perfusion was resumed, and media samples were collected in duplicate at $t = 0$ h, 1 h, 2 h, 7 h, and 23 h. Media ammonia levels were quantified on a CEDEX BioHT, and duplicate frozen samples were assayed in parallel to measure urea produced over time (Sigma, MAK0061KT).

**Acute blood perfusion studies.** For the in vitro blood perfusion studies, each BEL was connected to a circuit comprised of silicone tubing, a pressure transducer (Deltran, DPT-100), and a peristaltic pump (Cole-Palmer, 07522-20). Freshly collected, heparinized porcine blood was warmed to 37 °C and the activated clotting time (ACT) was measured (ITC, Hemochron Response). A solution of protamine sulfate was then gradually added to the blood to neutralize the heparin until an ACT of 170-220 was reached. 2 L of blood was introduced into the circuit and perfused through the BEL construct at an initial flow rate of 300 mL/min, and then immediately switched to pressure-dependent flow control targeting a constant pressure of 12 mmHg. Flow rates and pressures were recorded over 60 minutes of blood perfusion.

In vivo acute blood studies were performed using domestic swine weighing 80-100 kg after approval by the Institutional Animal Care and Use Committee (IACUC) at American Preclinical Services (Coon Rapids, MN). Animals were heparinized to a target ACT of 225 s. Recipient vessels: portal vein and iIVC, were cannulated using a 28 F single stage venous cannula (Medtronic). BELs were connected to portal venous blood flow using PVC (LivaNova) tubing and ¼" luer-lock connectors to achieve functional end-to-side anastomoses between the grafts' and recipients' portal veins and venae cavae. Flow rates were measured using a ½" ultrasonic flow probe (ME 11PXL) connected to a controller box (TS410; Transonic Systems Inc, Ithaca, NY, USA) and recoded manually. Flow through the BELs was visualized via venogram. Isovue contrast was injected directly into the perfusion loop upstream of the BEL and images were collected using an OEC 9900 Elite mobile C-arm (GE Healthcare).

**Heterotopic BEL implantation and portocaval shunt surgeries**

*ICP probe placement, portocaval shunt and liver implantation procedure.* All animal experiments were performed in accordance with the IACUC at Mayo Clinic (Rochester, MN) and American Preclinical Services (Coon Rapids, MN) and all experiments herein were performed in accordance with the guidelines and regulations of the committee. Twenty-eight to 36 kg domestic white swine were procured from a local USDA-certified (class A) vendor and blood typed for type AO or A via PCR on buccal swab samples (Zoologix Inc, Chatsworth, CA, USA).

For all animal implant procedures described below, anesthesia was induced via intramuscular injection of telazol (0.5 mg/Kg) and xylazine (0.2 mg/Kg). IV access was established for fluid resuscitation with 1 L 0.9% NaCl and administration of cefazolin 1 g for surgical prophylaxis. Extended-release opiate analgesia was provided. After endotracheal intubation, ventilation was maintained to achieve end-tidal $CO_2$ of 35-40 torr. Anesthesia was maintained with inhaled isoflurane 1-3%.

For ICP probe placement, animals were fasted 16 h prior to the procedure, but allowed water *ad lib.* A scalp flap was elevated, and a 4 mm Burr hole was drilled through the frontal bone 1.5 cm lateral to midline and 1 cm superior to the superior orbital foramen. The dura mater was punctured bluntly, and a transdermal telemetric intracranial pressure monitor (Raumedic, Helmbrechts, Germany) was introduced into the subdural space. The scalp flap was closed over the monitor and

the animal was allowed to recover for at least five days to allow for local swelling to subside and to observe for signs of wound infection which would result in elective euthanasia and removal from the study.

The portocaval shunt procedure was adapted from one described by Lee et al.[20]. Animals were transitioned to a soft food diet of Ensure and canned dog food (Hills Digestive Care a/d) three days prior to surgery and then fasted 16 h prior to the procedure, with access only to water *ad lib.* Anesthesia was induced via intramuscular injection of telazol (3.5-5.5 mg/kg) and xylazine (1.5–3.5 mg/kg). IV administration of 0.9% NaCl was used as necessary for fluid resuscitation and cefazolin 1 g for surgical prophylaxis. Ketamine (~2 mg/kg/h), midazolam (~0.6 mg/kg h) and fentanyl (~0.004 mg/kg/h) were used as necessary as adjuncts. Five hundred mg solumedrol IV was given intravenously for induction immunosuppression. A bladder catheter was placed. After endotracheal intubation, ventilation was maintained to achieve end-tidal $CO_2$ of 35–40 Torr. Anesthesia was maintained with inhaled isoflurane 0–5%.

A midline laparotomy was performed, and a self-retaining retractor was placed. Splenectomy was performed via hilar ligation and division taking care to remove any splenules and preserve the tail of the pancreas. All ligaments around the liver were taken down and any aberrant vasculature was ligated and divided. A complete hepatoduodenal ligament dissection was performed and the HAs, PV and CBD were isolated. The iIVC was mobilized inferiorly to the level of the right renal vein taking care to preserve large local lymphatics. The caudate lobe was devascularized with aggressive parenchymal compression using a running, locking 3-0 PDS suture until cut parenchymal edges did not demonstrate active bleeding.

In the control group, a direct portocaval anastomosis was performed. The animal was heparinized to a goal ACT of 170–225 s. The iIVC and the PV were partially clamped, and a side-to-side anastomosis 1 cm in diameter was performed. After ensuring patency, acute ischemic liver failure was induced by ligation and division of the HAs, PV upstream from the anastomosis and CBD. This represented time zero.

In the experimental group, a BEL graft seeded with HUVECs and porcine hepatocytes as described above was placed as opposed to a direct portocaval shunt. To reduce portal hypertension from the expected small-for-size syndrome, a 4 mm polytetrafluoroethylene (PTFE) shunt was anastomosed end-to-side between the recipient's PV and iIVC. Patency was shown by an increase in PV flow with conduit clamping as measured by an ultrasonic perivascular flow module (Transonic Systems Inc, Ithaca, NY, USA). Then, two 8 mm diameter, ringed PTFE prosthetic vascular grafts (W. L. Gore and Associates, Newark, DE, USA) were anastomosed to the portal vein and iIVC of the BEL using running 6-0 prolene suture to bolster the anastomoses and provide additional length if needed based on the animals' anatomy (Fig. S4). Preservation solution was flushed from the liver graft using 0.9% NaCl. The liver graft was liberally coated with Tisseel aerosolized fibrin sealant (Baxter Healthcare Co., Deerfield, IL, USA) to provide a physiologic pseudocapsule and permit handling and retraction as needed during implantation and to provide a bolster against graft fracture and resultant uncontrollable hemorrhage secondary to overinflation from portal hypertension. The liver graft was introduced into the abdomen and placed in the auxiliary position inferior to the native liver directly anterior to the right adrenal gland. The animal was heparinized to a goal ACT of 170-225 sec. The recipient iIVC and portal vein were partially clamped, and end-to-side anastomoses were performed to the BEL's PTFE vascular grafts with inflow consisting of the native portal vein flowing to the BEL portal vein and outflow consisting of the BEL's iIVC flowing into the recipient's iIVC. The graft's vasculature was filled with 0.9% NaCl through iIVC, and the graft was reperfused by unclamping the inflow and allowing antegrade blood flow to vent through the BEL's sIVC prior to ligation of the BEL sIVC and unclamping of the outflow anastomosis. PV inflow to the BEL was measured with an ultrasonic perivascular flow module to ensure flow above 120 mL/min. If blood flow was below this value, then the PTFE shunt was banded or ligated to increase flow as necessary. Hemostasis was achieved with suture ligation or application of topical fibrin sealant). Whole blood transfusion of type-A blood up to 1000 mL was used as needed to correct for blood loss or hemodynamic instability. Acute ischemic liver failure was induced by ligation and division of the hepatic arteries, PV distal to the anastomosis and common bile duct. This represented time zero.

Once the animals were vitally stable and hemostasis was achieved, an abdominal drain was placed in the surgical field and connected to bulb suction. Effluent volume and character were recorded throughout the remainder of the study. In the case of two animals, the ultrasonic flow probe was left on the graft portal vein to monitor flow during the monitoring period. The abdomen was closed in multiple layers.

**Monitoring and resuscitation.** Following surgery, a strict, standardized monitoring and resuscitation protocol was utilized which involved hourly monitoring of vital signs, fluid output, hemodynamic parameters and ICP until death. Sedation was maintained with isoflurane 1–3% inhaled. Crystalloid resuscitation of up to 300 mL/h and administration of phenylephrine up to 1 mcg/Kg/min titrating to a mean arterial pressure (MAP) of 50 mmHg was permitted. Five percent dextrose solution was added to crystalloid maintenance fluid to maintain blood glucose of 60-120 mg/dL. The target body temperature of 37 °C was maintained with a heating blanket. Endpoint was achieved when the animal had two consecutive

hourly measurements of MAP < 30 mmHg or ICP > 20 mmHg and was euthanized via pentobarbital overdose.

**Animal imaging**. Experimental animals were scanned via computer-assisted tomography (CT) of the abdomen and pelvis on a SOMATOM Definition VA44A CT scanner (Siemens AG, Munich, Germany) post-operatively. 60–90 mL of IV Optiray 350 Ioversol 350 mg/mL was administered at a contrast:saline ratio of 80:20 immediately prior to scanning. Five scans were taken every 15 s to ensure graft patency and successful ligation and division of all inflow vessels to the native liver in situ.

**Biochemical analysis**. Blood samples were obtained at time zero and every four hours following induction of acute ischemic liver failure (Supplementary Fig. S2). Blood ammonia ($NH_3$), albumin (Alb), creatinine (Cre), creatine kinase, aspartate aminotransferase (AST), alanine aminotransferase (ALT), total bilirubin (tBil), gamma-glutamyl transferase (GGT), lactate dehydrogenase (LDH) and potassium levels were measured. Blood glucose (Glu) and prothrombin time (PT/INR) were determined using an automatic point-of-care biochemical analyzer (Abbot Point of Care Inc., Abbott Park, IL, USA).

**Statistics and reproducibility**. Independently decellularized, seeded and cultured BEL constructs were defined as biological replicates for purpose of all data analyses conducted in this study. Computed mean values and standard deviations were shown where applicable. All experiments were repeated at least twice to confirm the reproducibility of the results. All experimental groups were comprised of at least two independent biological replicates.

**Reporting summary**. Further information on research design is available in the Nature Research Reporting Summary linked to this article.

## Data availability
Source data for graphs and charts in the main figures are present as Supplementary Data 1 and any remaining information can be obtained from the corresponding author upon reasonable request.

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

## Author contributions
B.D.A., D.S.D., J.J.R., S.L.N. designed the study; B.D.A., E.D.N., A.A.K., A.M., B.G.S., D.S.D., T.W.G., J.J.R., S.L.N. analyzed the data; B.G.S. seeded and cultured the BELs; B.G.S., T.N. and N.P. performed the histological staining; B.D.A. and A.M. performed the bioreactor media assays; E.D.N., D.J.J., B.P.A., S.L.N. performed the surgical procedures; A.A.K. performed the ex vivo blood perfusion assays; A.R.S. performed the flow cytometry assays; B.D.A. and T.N. drafted the figures; B.D.A., E.D.N., D.S.D., T.W.G., J.J.R., and S.L.N. wrote the manuscript

## Competing interests
The authors declare the following competing interests: B.D.A., A.A.K., A.M., B.G.S., A.R.S., V.L.N., R.N.P., T.W.G., D.S.D., and J.J.R. are employees of Miromatrix Inc. Miromatrix Inc. is a publicly funded company and owns the exclusive patent rights for the perfusion decellularization and recellularization technologies utilized in this study. The remaining authors declare no competing interests.
