## [Transparent Peer Review File · Communications Biology]

Reviewers' comments:

Reviewer #1 (Remarks to the Author):

In this manuscript, the authors reported ex vivo and in vivo experiments of decell/recellularized porcine liver with primary porcine hepatocytes and endothelial cells based on their previous work, in which they had already reported the similar ex vivo and in vivo experiments but just with endothelial cells.

Therefore, significance of this manuscript should be the hepatic function of the engineered liver, especially how close it is to native liver graft. Since the recellularized liver which was applied for their experiments eventually contains a large amount of primary hepatocytes, they are required to show the hepatic function precisely not only by NH₃ clearance and urea production but by Cyp450 activity or gene expression in ex vivo evaluation or more appropriate histological study including sinusoid construction or alignment of the infused hepatocytes.

In addition, the one of the most unconvinced part of this manuscript is that in vivo model they applied was too acute and technically complicated, which could even minimize the strength of the meaning of this manuscript.

Since, they have already showed 10 days after the implantation of the endothelialized scaffold in prior study, they are at least required to explain why they removed the graft at just 48 hours. We would like to know if there were any difficulties to apply semi-acute model or chronic model, such as elegant long-term liver disfunction model "FAH pig", which actually the authors have investigated in 2011.

Major comments:

In figure2, please provide the picture of the entire surface of the recellularized liver not only just medium magnification of part it, which helps to recognize the efficacy of recellularization process of the hepatocytes.

In figure2, please provide Cyp450 activity or gene expression of the engineered liver and better to compare them to healthy pig liver.

In figure2, also please provide the pictures of immune histochemistry with high magnification to show the alignment and function of the infused cells.

In figure3, it should be difficult to remove the native liver in pigs because of its anatomical issue, but still there is a concern that neo-vascularization could come into the ischemic native liver and raised some hepatic function. The authors are required to avoid this possible experimental bias by the evaluation of the remnant livers.

In figure3, the in vivo experiment, which was terminated at 48 hours, is too short to see the survival or hepatic function of the engineered liver. Please provide data for at least 10 days as the prior experiment.

In figure3, the number of histological pictures are very few and resolution is too low to see what happens in the engineered liver after implantation. Please provide more precise data of histological study, and also it is good to show the data of NH₃ clearance or Urea production after implantation comparing to the one from pre-implantation.

In figure3, the readers may concern how the bile production in the implanted bioengineered liver was. It was just 48 hours but healthy hepatocytes may start producing bile in vivo. Please provide any data like histology or gene expression to show how it was.

Reviewer #2 (Remarks to the Author):

This manuscript is exceptionally well-written and convincing in its conclusion. It represents an immense amount of work. The authors are well published in the field, and have increased their contributions with this work.

Large scale hybrid organs utilizing a decellularized porcine scaffold seeded with HUVECS and porcine

endothelial cells were successfully transplanted heterotopically.

Some unique features to their approach include:

Maximizing the success of graft creation by: utilizing a clean lab, sterilizing the decellularized construct and "proving" graft quality with the glucose assay, ICP monitoring, creation of a neocapsule with fibrin glue, and implantation with portal caval shunts.

Their overall approach shares the same basic limitation of all scaffold groups: complete regeneration of a functional organ and figuring out which cell types to use. It is surprising that a pig-scaffold based graft seeded with human endothelial cells transplanted into a pig would have different "long" term outcomes than a pig liver transplanted into a human (minimal long term survival in historical controls.) Have the authors perfused identical grafts with human blood? Would you expect even better survival?

The authors point out that repopulation with hepatocytes alone will not be adequate for parenchymal restoration. Are there ongoing plans to address the biliary system? I understand that the commercial nature of this project may limit this disclosure.

John LaMattina

Reviewer #3 (Remarks to the Author):

The authors have constructed engineered liver grafts for heterotopic transplantation in a porcine model. Following transplantation, a maximal 48 hrs of survival time was reported. While the study demonstrated the therapeutic potential of the bioengineered liver grafts, I personally feel it is not adding more information to the field than what has been previously reported, such as Ko et al., *Biomaterials*, 40:72-79, 2015; Mao et al., *Int J nTransplant Res Med* 3:031, 2017. In particular, the study done by Mao et al., has shown a in vivo survival time of at least 3 days.

The study would be more publishable if the authors can prolong the survival time of the hosts.

Reviewers' comments:

Reviewer #1:

In this manuscript, the authors reported ex vivo and in vivo experiments of decell/recellularized porcine liver with primary porcine hepatocytes and endothelial cells based on their previous work, in which they had already reported the similar ex vivo and in vivo experiments but just with endothelial cells.

Therefore, significance of this manuscript should be the hepatic function of the engineered liver, especially how close it is to native liver graft. Since the recellularized liver which was applied for their experiments eventually contains a large amount of primary hepatocytes, they are required to show the hepatic function precisely not only by NH₃ clearance and urea production but by Cyp450 activity or gene expression in ex vivo evaluation or more appropriate histological study including sinusoid construction or alignment of the infused hepatocytes.

We thank the reviewer for their comment, and we have added immunohistochemical staining of sections of the bioengineered liver that show Cyp3A4 activity by the hepatocytes prior to implantation and at the time of explant (Figure 4 h-i). The number and intensity of positive Cyp3A4 staining tends to increase after BEL implantation and native blood perfusion. We highlight the limitations of the study but believe this is a significant advancement to liver bioengineering. Given the cost of large animal studies, it is challenging to repeat these studies to obtain gene expression data.

In addition, the one of the most unconvinced part of this manuscript is that in vivo model they applied was too acute and technically complicated, which could even minimize the strength of the meaning of this manuscript.

Since, they have already showed 10 days after the implantation of the endothelialized scaffold in prior study, they are at least required to explain why they removed the graft at just 48 hours. We would like to know if there were any difficulties to apply semi-acute model or chronic model, such as elegant long-term liver disfunction model “FAH pig”, which actually the authors have investigated in 2011.

We thank the reviewer and understand that lack of an ideal large animal model for these studies. The previously published study with 10 days survival after implantation of an endothelialized liver scaffold was intended only to show long term patency of the graft with human endothelial cells, which was necessary before attempting to add hepatocytes. As the native liver was not compromised in the previously published model, the endpoint of the study was based upon immune mediated rejection that led to a loss of patency, not due to acute liver failure.

The present work represents an initial step to show that hepatocytes seeded into a decellularized liver scaffold can be anastomosed, perfused with native blood and restore some function in an acute and, as the reviewer points out, technically challenging acute liver failure model. Further, due to the heterotopic nature of the model and the presence of the native liver, there was limited

space for the bioengineered liver. Given these challenges, we believe this work shows proof of concept that the hepatocytes seeded into a decellularized liver can show initial hepatic function.

We must emphasize that our goal was to test the graft in an acute liver failure model. Using the FAH-knockout pig would be an elegant way to test the graft in a chronic liver failure model, which may be a future direction.

Major comments:

In figure2, please provide the picture of the entire surface of the recellularized liver not only just medium magnification of part it, which helps to recognize the efficacy of recellularization process of the hepatocytes.

We thank the reviewer for this suggestion. A photograph of a representative recellularized bioengineered liver is provided. Overall, the number of hepatocytes seeded was relatively low compared to the native liver due to the size of graft that could be placed within the recipient due to the presence of native liver. Our seeding method results in engraftment in particular regions of the graft and localized in particular lobules rather than diffusely throughout the BEL. We believe this method further promotes hepatocyte function to increase cell to cell interactions. This description is included in the Results section, and further thoughts regarding scale-up of the seeding is included in the Discussion.

In figure2, please provide Cyp450 activity or gene expression of the engineered liver and better to compare them to healthy pig liver.

We thank the reviewer for this suggestion and a histologic image showing positive immunostaining for Cyp450 has been added to Figure 4.

In figure2, also please provide the pictures of immune histochemistry with high magnification to show the alignment and function of the infused cells.

We thank the reviewer and have added additional pre-implantation images were added to Figure 4 that also demonstrate alignment pre and post implantation. In addition, it was an increased expression of CYP3A4 and Albumin was post implantation was noted.

In figure3, it should be difficult to remove the native liver in pigs because of its anatomical issue, but still there is a concern that neo-vascularization could come into the ischemic native liver and raised some hepatic function. The authors are required to avoid this possible experimental bias by the evaluation of the remnant livers.

We thank the reviewer for this point but respectfully believe that it is not anatomically possible for neo-vascularization to reestablish arterial perfusion through a ligated and divided hepatic plate in the 48 hour time frame. If the plate was not totally divided, there could be accessory hepatic arteries which were not divided and could dilate to provide additional arterial perfusion. To confirm this point, Figure 4 includes reconstructions from CT scans showing almost complete

absence of vascularization of the native liver immediately post-operatively. While the Ct data provides the most convincing support for lack of vascular supply, Figure S5 was added to provide an example of the an explanted devitalized native liver.

In figure3, the in vivo experiment, which was terminated at 48 hours, is too short to see the survival or hepatic function of the engineered liver. Please provide data for at least 10 days as the prior experiment.

We thank the reviewer for their comment and respectfully disagree that 48 hours is too short given the limitation of large animal models. As stated above, the previously published study with 10 days survival after implantation of an endothelialized liver scaffold into a pig with a functioning native liver was intended only to show long term patency of the graft, which was necessary before attempting to add hepatocytes. The native liver was intact during the entirety of the study; therefore we do not believe that comparing the length of survival in this study to the previously published work is appropriate at this point in time as we are studying acute liver failure in this paper and it wasn't intended to be a recovery model.

The present work represents a critical step to show that hepatocytes seeded into a clinically – scaled, decellularized scaffold can restore some function in an acute and technically challenging liver failure model. Due to the heterotopic nature of the model and the presence of the native liver, there was limited space for the bioengineered liver, which limited number of hepatocytes that could be added to the bioengineered organ to replace liver function. Still this work shows better initial function, on a more clinically relevant scale than any previously published work. Achieving survival of 10 days and longer will be the goal of future experiments that will require a larger bioengineered liver with more hepatocytes. We continue to pursue that goal.

In figure3, the number of histological pictures is very few and resolution is too low to see what happens in the engineered liver after implantation. Please provide more precise data of histological study, and also it is good to show the data of NH3 clearance or Urea production after implantation comparing to the one from pre-implantation.

We thank the reviewer for their comment, and we have added pre and post implantation immunohistochemical data to Figure 4 for Albumin and CYP3A.

The blood ammonia level shows consistently low ammonia concentration in the blood for each animal at the time that surgery was initiated to create the acute liver failure model. This data is representative of baseline ammonia levels.

In figure3, the readers may concern how the bile production in the implanted bioengineered liver was. It was just 48 hours but healthy hepatocytes may start producing bile in vivo. Please provide any data like histology or gene expression to show how it was.

We thank the reviewer for their comment and acknowledge that this would be a good follow-up study. Unfortunately, in the current study specimens for gene expression analysis were not secured during the study, and we have been unable to identify antibodies to indicate bile

production in vivo for pigs, either due to a lack of cross species recognition by human antibodies, due to the lack of bile production at this time point, or due to the lack of concentration of bile due to the absence of function bile ducts. Given the complex nature and cost of large animal studies it isn't possible to provide this data, but it will be a focus on the next studies in a recovery model.

Reviewer #2:

This manuscript is exceptionally well-written and convincing in its conclusion. It represents an immense amount of work. The authors are well published in the field, and have increased their contributions with this work.

Large scale hybrid organs utilizing a decellularized porcine scaffold seeded with HUVECS and porcine endothelial cells were successfully transplanted heterotopically.

Some unique features to their approach include:

Maximizing the success of graft creation by: utilizing a clean lab, sterilizing the decellularized construct and "proving" graft quality with the glucose assay, ICP monitoring, creation of a neocapsule with fibrin glue, and implantation with portal caval shunts.

Their overall approach shares the same basic limitation of all scaffold groups: complete regeneration of a functional organ and figuring out which cell types to use. It is surprising that a pig-scaffold based graft seeded with human endothelial cells transplanted into a pig would have different "long" term outcomes than a pig liver transplanted into a human (minimal long term survival in historical controls.) Have the authors perfused identical grafts with human blood? Would you expect even better survival?

We thank the reviewer for their comments and we agree that persistent blood flow through an engineered liver reendothelialized with HUVECs should be predictive of greater longevity after future implantation in humans. Immune acceptance of the HUVECs should be increased when implanted into a human relative to the xenorejection observed when HUVECs are implanted into a pig. While we know that a pig liver into a human is immediately rejected, the converse study hasn't been done to our knowledge and we know certain antigens such as alpha-gal would not be a factor in the human into a pig model. We would like to note, that our ultimate goal for a therapeutic liver would be 100% human cell recellularization. The current study helps support the advancement of this work.

We have not perfused identical grafts with human blood at this point but we are planning to do it in the future. Our in vitro blood loops tend to run for relatively short period of time (approximately 1 hour), and the system is not intended to preserve the quality of the blood over extended periods of time. Therefore, we would not predict different results compared to perfusing porcine blood in vitro.

The authors point out that repopulation with hepatocytes alone will not be adequate for parenchymal restoration. Are there ongoing plans to address the biliary system? I understand that the commercial nature of this project may limit this disclosure.

We thank the reviewer for their comment and the discussion about the limitation of only seeding hepatocytes was related to future goals of a fully function liver for treatment of chronic liver failure. However, in this manuscript the goal was treatment of acute liver failure, not chronic liver failure. We believe bile production and elimination is not a goal of care in acute liver failure like ammonia clearance is for acute therapy.

With that said, we are actively working on the biliary system at this time and look forward to reporting on that work in a future study with a focus on treatment of chronic liver failure in a recovery model. The present work provides the basis for an acute liver therapy pathway while also supporting future work on the chronic model.

Reviewer #3:

The authors have constructed engineered liver grafts for heterotopic transplantation in a porcine model. Following transplantation, a maximal 48 hrs of survival time was reported. While the study demonstrated the therapeutic potential of the bioengineered liver grafts, I personally feel it is not adding more information to the field than what has been previously reported, such as Ko et al., *Biomaterials*, 40:72-79, 2015; Mao et al., *Int J Transplant Res Med* 3:031, 2017. In particular, the study done by Mao et al., has shown a in vivo survival time of at least 3 days.

The study would be more publishable if the authors can prolong the survival time of the hosts.

We thank the reviewer and would like to note that Ko et al. and Mao et al. only report on bioengineered livers seeded with endothelial cells to evaluate perfusion in the graft. Our group has subsequently published a longer term study (Shaheen et al., *Nat Biomed Eng*, 2019) that extended the survival of an endothelial cell seeded graft to 10 days. The major difference between the present study and all three of these earlier studies is the addition of hepatocytes to evaluate the potential for the bioengineered liver to restore some function in the setting of a compromised native liver. Furthermore, as this was the initial attempt to rescue a large animal with compromised liver function, the animals were sedated for the duration of the study, in contrast to the previous studies in which the animals were resuscitated.

In total, the conditions in the present study were quite different from those presented in the previous studies, which makes it inappropriate to utilize those studies as a direct benchmark for success in the present study. Unlike the previous study where the native liver was intact, these animals were effectively anhepatic and while also experiencing a major transplant surgery (Figure S4). Despite the complexities of this model, the present study is the first to report suggested improvements in liver function based upon implantation of a bioengineered liver, and forms the foundation for future studies.

Reviewers' comments:

Reviewer #1 (Remarks to the Author):

I could not see the improvement on the pictures which should have high resolution of immunohistochemical study for pre-and post-implant liver functions in figure2 and 4. Most of the positive stain of the cells did not match to DAPI stain positive cells, especially albumin and CYP stain of the post-implant tissues. I will ask to the authors again to provide better pictures in which readers can see albumin and CYP positive cells clearly.

I did not understand that although the authors claimed that they had planned to show hepatocyte/liver function under this acute liver failure model, they did not take any samples for gene expression analysis which is one of the fundamental ways to prove hepatocyte/liver function. I assume that the tissue which was remaining after that implantation was extremely small, which was not feasible for additional analysis. The authors are able to avoid this question by showing morphological pictures of post-implant bio-engineered liver tissue.

I asked the authors to provide the picture of entire surface of bioengineered liver, which strongly help the readers to see how much space had been occupied by the infused hepatocytes and how close it was to the healthy liver however, the picture was not provided.

I do not understand yet that the reason of why the authors chose this extremely short-term experiment. Since, they did not show enough data to demonstrate the efficacy of this bioengineered liver in treating acute liver failure by expressing liver function as mentioned above. Since they claimed that they had at least 10 days period to show the efficacy of this bioengineered liver based on their prior study, they could have used more relevant model which could show longer observation period such as CCl₄ or acetaminophen induction. I assume that if the bioengineered liver had any fundamental difficulty to show longer function. To avoid this question, the authors should show more precise data of hepatocyte/liver function such as AST/ALT/g-GTP/ALP/PT% and morphology of the graft after implantation, additionally sinusoidal structure of the graft.

Reviewer #2 (Remarks to the Author):

The authors have adequately addressed my earlier concerns.

As a service to authors, Springer Nature provides authors with the ability to transfer a manuscript that one journal cannot offer to publish to another journal, without the author having to upload the manuscript data again. To transfer your manuscript to another Springer Nature journal using this service, please click on <https://mts-commsbio.nature.com/cgi-bin/main.plex?el=A2Cx7BJL4B2VKb6X2A9ftdqjygtNT1dNc1jV4kyxU4wZ>

Note that any decision to opt in to In Review at the original journal is not sent to the receiving journal on transfer. You can opt in to [In Review](https://www.nature.com/nature-research/for-authors/in-review) at receiving journals that support this service by choosing to modify your manuscript on transfer. In Review is available for primary research manuscript types only.

Concern 1 - *I could not see the improvement on the pictures which should have high resolution of immunohistochemical study for pre-and post-implant liver functions in figures 2 and 4.*

Response: The image quality of both Figures has been improved. Also, Figure 4 has been updated to correct a typo so that Fig. 4 (h,j,l) refer to pre-implant histology and Fig 4. (l,K,m) refer to post implant histology. The resolution and magnification of the images are now increased (200 um bars added) resulting in an increased resolution for CYP3A4 to address reviewer's concerns. Supplementary Figure S7 was added showing higher magnification images to address the concerns also.

Concern 2 - Most of the positive stain of the cells did not match to DAPI stain positive cells, especially albumin and CYP stain of the post-implant tissues. I will ask to the authors again to provide better pictures in which readers can see albumin and CYP positive cells clearly.

Response: Supplement Figure S7 was added to address the reviewer's concerns. Figure S7 contains higher magnification images to address the reviewer's concerns.

Concern 3 - I did not understand that although the authors claimed that they had planned to show hepatocyte/liver function under this acute liver failure model, they did not take any samples for gene expression analysis which is one of the fundamental ways to prove hepatocyte/liver function. I assume that the tissue which was remaining after that implantation was extremely small, which was not feasible for additional analysis. The authors are able to avoid this question by showing morphological pictures of post-implant bio-engineered liver tissue.

Response: The bioengineered liver graft retains its shape. To address the reviewer's comment a representative image of the explanted bioengineered liver was added as Supplementary Figure S6.

Concern 4 - I asked the authors to provide the picture of entire surface of bioengineered liver, which strongly help the readers to see how much space had been occupied by the infused hepatocytes and how close it was to the healthy liver however, the picture was not provided.

Response: Examples of the bioengineered liver and the associated implantation were previously added in Fig. 2 and Supplementary Figure S4. To further address the reviewer, Supplementary Figure S5 has been added to provide clear visualization of the surface of the bioengineered liver.

Concern 4 - I do not understand yet why the authors chose this extremely short-term experiment. Since, they did not show enough data to demonstrate the efficacy of this bioengineered liver in treating acute liver failure by expressing liver function as mentioned above. Since they claimed that they had at least 10 days period to show the efficacy of this bioengineered liver based on their prior study, they could have used more relevant model which could show longer observation period such as CCl4 or acetaminophen induction. I assume that if the bioengineered liver had any fundamental difficulty to show longer function. To avoid this question, the authors should show more precise data of hepatocyte/liver function such as AST/ALT/g-GTP/ALP/PT% and morphology of the graft after implantation, additionally sinusoidal structure of the graft.

Response: We considered the possibility of other large animal models (CCl4 or acetaminophen induction) for testing of the BELS grafts in vivo, however the reproducibility of the toxicity models is variable in pigs. Therefore, because of its superior reproducibility a surgical model of acute liver failure was utilized in pigs. We agree with the reviewer regarding the challenge of demonstrating biochemical function of a bioengineered liver implanted in an auxiliary heterotopic position due to the persistence of liver function by the native remnant liver. For this reason, liver function was measured during ex vivo perfusion of the bioreactor as reported in Figure 2. Biochemical data obtained in vivo after graft implantation was added to a prior revision of the manuscript as Supplementary Figure S2. Of note, blood levels of AST/ALT/g-GTP/ALP are more recognized as markers of hepatocyte injury than true measures of liver function. Rather, lowering blood levels of bilirubin and ammonia and production of albumin are better measures of liver function *in vivo*, and these markers were improved in our treatment animals.

REVIEWERS' COMMENTS:

Reviewer #4 (Remarks to the Author):

The authors demonstrate a method for repopulation of decellularized porcine livers with both HUVECs and porcine hepatocytes building on their previous report of HUVEC-only repopulation. The authors have carried out key assays of hepatocyte function including albumin production and ammonia metabolism/urea synthesis. While BEL hepatocytes appear functional for 48 hours in vitro, hepatocytes retain partial function in vivo following heterotopic BEL transplantation with low bilirubin and ammonia levels but also decreasing albumin levels on par with the portocaval shunt control. Indeed only one of three transplanted BELs was able to sustain animal life to the study endpoint, which may also be related to the experimental caveats the authors describe in detail. Nevertheless generation of transplantable BELs with viable and even partially functional hepatocytes is an achievement and constitutes an important advance in the effort to generate transplantable livers of a clinically relevant size for humans. These studies provide a foundation from which to further optimize this system and bring us closer to realizing this goal.

The authors have sufficiently characterized BEL function. They have included high resolution and increased magnification representative pre- and post-implant images of CYP3A4, FAH, albumin, and CD31. However, they do not show any post-implant LYVE1 or any images that would allow for assessment of post-implant sinusoidal structure. Since LYVE1 staining and sinusoidal structure is shown in the in vitro BELs, it would be nice to see how the transplanted BELs compare. While they do not show any gene expression, they do show biochemical data and staining indicative of transplanted BEL function. Images showing the surface of the liver after hepatocyte seeding and upon BEL explant have also been provided.

Minor comments:

The authors claim flow rates of 150-250 mL/min in their ex vivo blood perfusion model (Fig 3) but only show angiography data. The authors should show their flow rate data.

It's not clear how long BELs are cultured with hepatocytes before transplantation into their in vivo liver failure model. It would be helpful for the authors to clarify this point.

Please include concentrations of each culture media additive in the methods section.

Figure 2 legend panels are mislabeled.

Reviewers' comments:

Reviewer #4:

Since LYVE1 staining and sinusoidal structure is shown in the in vitro BELs, it would be nice to see how the transplanted BELs compare. While they do not show any gene expression, they do show biochemical data and staining indicative of transplanted BEL function.

We thank for the reviewer for pointing out our biochemical data and staining. At the request of the reviewer pre and post LYVE1 staining was added to the supplement in S8.

The authors claim flow rates of 150-250 mL/min in their ex vivo blood perfusion model (Fig 3) but only show angiography data. The authors should show their flow rate data.

The data was collected manually based on the flow rates after 30 minutes. We clarified this point in the results section. There isn't any additional data to show or place into a figure since they are individual timepoints.

It's not clear how long BELs are cultured with hepatocytes before transplantation into their in vivo liver failure model. It would be helpful for the authors to clarify this point.

We thank the reviewer for catching identifying this. Clarification that hepatocytes were seeded and cultured for 48 hours before transplantation was added to the results sections.

Please include concentrations of each culture media additive in the methods section.

Concentrations of media additives were added to the methods section

Figure 2 legend panels are mislabeled.

We thank the reviewer for noting this and the legend panels were corrected.